

# Absence of dynamical localization in interacting driven systems

David J. Luitz[1*], Yevgeny Bar Lev[2] and Achilleas Lazarides[3]

**1** Department of Physics, T42, Technische Universität München, James-Franck-Straße 1, D-85748 Garching, Germany
**2** Department of Chemistry, Columbia University, 3000 Broadway, New York, New York 10027, USA
**3** Max-Planck-Institut für Physik komplexer Systeme, 01187 Dresden, Germany

⋆ david.luitz@tum.de

## Abstract

Using a numerically exact method we study the stability of dynamical localization to the addition of interactions in a periodically driven isolated quantum system which conserves only the total number of particles. We find that while even infinitesimally small interactions destroy dynamical localization, for weak interactions density transport is significantly suppressed and is asymptotically diffusive, with a diffusion coefficient proportional to the interaction strength. For systems tuned away from the dynamical localization point, even slightly, transport is dramatically enhanced and within the largest accessible systems sizes a diffusive regime is only pronounced for sufficiently small detunings.

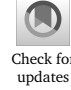

# 1   Introduction

The last decade has witnessed an increasing interest in isolated out-of-equilibrium (OOE) quantum systems both in theory and experiment. In particular, periodically-driven, or Floquet many-body systems have become one of the most active areas in condensed matter theory and many-body physics, due to the exciting possibilities of finding novel OOE phases of matter, without an analog in static systems.

Driven interacting systems typically heat up, until they approach a nonequilibrium steady-state (NESS) "locally identical" to a featureless state of maximal Gibbs entropy. That is, *local* observables cannot differentiate between the two states [1–3].

This "local identity" between the NESS and the maximal Gibbs entropy state is broken when conservation laws exist, for example in noninteracting systems [4, 5]. The systems still approach a NESS, but this NESS is "locally identical" to a state which maximizes the Gibbs entropy under *all* the constraints imposed by the conserved quantities. A large body of work has concentrated on "Floquet engineering" of such noninteracting systems, namely using periodic driving as a tool to create effective time-independent Hamiltonians of desirable but difficult to implement static Hamiltonians [6–8].

For interacting systems it is possible to avoid the featureless NESS by the addition of a quenched disorder. In the absence of driving, these systems are many-body localized (MBL) [9–16], and for sufficiently high driving frequencies they do not heat up indefinitely [15, 16]. Such systems were also shown to host nontrivial eigenstate order, corresponding to a new OOE phase of matter, colloquially dubbed a "time-crystal" [17] and subsequently studied in both experimental and theoretical studies [18–21].

In the present work we explore a different potential avenue to suppress the indefinite heating of interacting driven systems by starting with a noninteracting translationally invariant system which is dynamically localized [22–26] (see also the closely related study of a *disordered* system [27], which appeared while our work was in preparation). A different dynamical localization mechanism was studied in Ref. [28]. The special form of the driving term in dynamically localized systems leads to a frozen stroboscopic dynamics and therefore to disorder-free spatial localization. Whether adding interactions to such disorder-free localized systems results in heating, and what is the nature of the transport are the main questions we consider in our work. We would like to stress that the mechanism behind this localization, while of quantum nature, is *not* due to Anderson localization, as happens in related disorder-free kicked rotor systems which unfortunately are also dubbed "dynamically localized" [29–31]. The interacting versions of such systems were recently studied, but are not of direct relevance to our work [32, 33].

The structure of this work is as follows: We present the model in Section 2, briefly survey dynamical localization adapted to the many-body context in Section 3 and in Section 4 present our main results showing that interactions destroy dynamical localization and cause diffusive dynamics with a diffusion coefficient proportional to the interaction. We end with our conclusions, a brief discussion of the implications of our results for the locality of Floquet effective Hamiltonians and an outlook.

# 2   Model

In this work we consider a driven Hamiltonian of the form,

$$\hat{H} = \hat{H}_0 + f(t) \sum_{m=1}^{L} m\hat{n}_m, \tag{1}$$

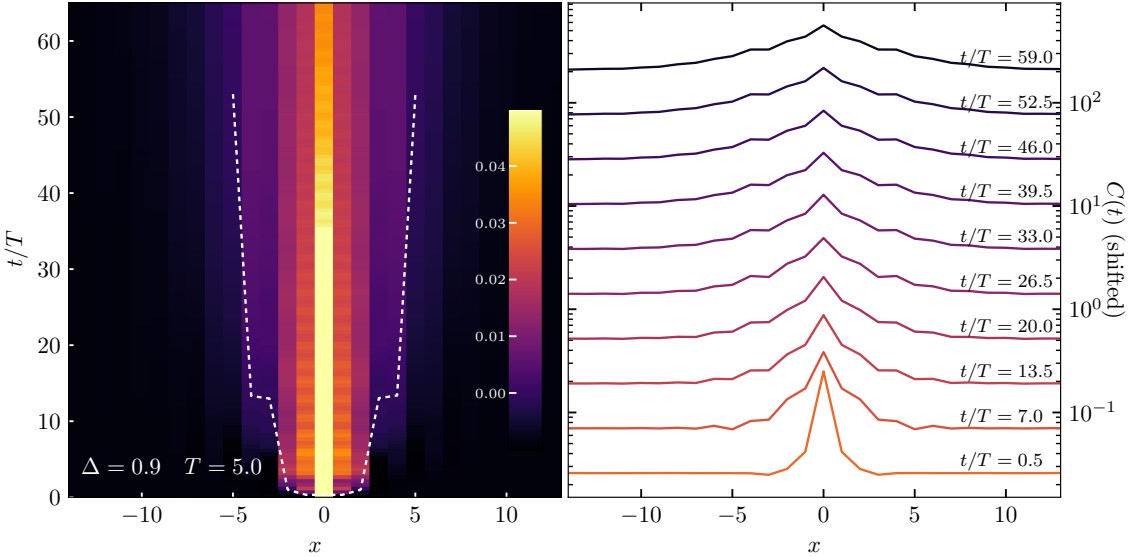

Figure 1: Spreading of a density excitation. The left panel shows the contour plot of the excitation quantified by $C_x(t)$ as a function of time $t$ and space $x$ (cf. Eq. (19)), with more intense colors representing larger values. The white line is an isoline set at $C_x(t) = 0$. The right panel shows cuts through the excitation profile at different times (origin offset for readability). The parameters used in this simulation are: $L = 29$, $T = 5$, $\Delta = 0.9$ and the amplitude of the drive is set to the dynamical localization point $A_0 = 4\pi/T$ (23).

where the static Hamiltonian is given by

$$\hat{H}_0 = -\frac{J}{2} \sum_{m=1}^{L-1} \left( \hat{c}_m^\dagger \hat{c}_{m+1} + \hat{c}_{m+1}^\dagger \hat{c}_m \right) + \Delta \sum_{m=1}^{L-1} \hat{n}_m \hat{n}_{m+1}. \tag{2}$$

Here $L$ is the length of the lattice, $\hat{c}_m^\dagger$ creates a spinless fermion at site $m$, $\hat{n}_m$ is the number operator, $J$ is the hopping and $U$ is the interaction strength, and the driving protocol is

$$f(t) = \begin{cases} -A & -\frac{T}{2} \le t < 0 \quad (\mathrm{mod}\, T) \\ A & 0 \le t < \frac{T}{2} \quad (\mathrm{mod}\, T), \end{cases} \tag{3}$$

where $T = 2\pi/\omega$ is the period of the driving, $\omega$ is the driving frequency and $A$ is the amplitude of the drive. This model corresponds to a constant potential gradient with a strength proportional to $A$, the sign of which is flipped every half period.

## 3 Dynamical localization

For $\Delta = 0$, the model (1) is noninteracting and exactly solvable [22, 23]. For specific ratios of $A/\omega$ it exhibits dynamical localization, namely, an initially localized density excitation *does not spread over time*. For convenience of the reader in this Section we adapt the derivation of dynamical localization of Ref. [23] to the many-body setting. For this purpose it is instructive to perform a *time-dependent* unitary transformation which eliminates the drive. This can be always achieved, since setting $|\psi\rangle = V(t)|\phi\rangle$ and using the Schrödinger equation, $i\partial_t |\psi\rangle = \hat{H} |\psi\rangle$ it is easy to show that $|\phi\rangle$ satisfies $i\partial_t |\phi\rangle = \tilde{H}(t)|\phi\rangle$, with the transformed Hamiltonian

$$\tilde{H}(t) = V^\dagger(t) H(t) V(t) - i V^\dagger(t) \partial_t V(t). \tag{4}$$

Therefore if the original Hamiltonian is of the form

$$\hat{H} = \hat{H}_0 + f(t)\hat{H}_1, \tag{5}$$

the second term can be eliminated by choosing

$$\hat{V} = e^{-i\mathscr{A}(t)\hat{H}_1}, \tag{6}$$

where $\mathscr{A}(t) = \int_0^t d\bar{t}\, f(\bar{t})$. The transformed Hamiltonian is then given by

$$\tilde{H} = e^{i\mathscr{A}(t)\hat{H}_1}\hat{H}_0 e^{-i\mathscr{A}(t)\hat{H}_1}. \tag{7}$$

While the original driving term was eliminated, it appears that nothing was gained since the transformed Hamiltonian is still time-dependent. For a spatially uniform and temporally periodic force on the system, namely

$$\hat{H}_1 = \sum_k k\hat{n}_k, \tag{8}$$

further progress can be made. For such a drive the destruction and creation operators transform as,

$$\hat{a}_m = e^{-i\mathscr{A}(t)m}\hat{c}_m \qquad \hat{a}_m^\dagger = e^{i\mathscr{A}(t)m}\hat{c}_m^\dagger, \tag{9}$$

and the number operators are invariant under this transformation, $\hat{n}_m = \hat{c}_m^\dagger\hat{c}_m = \hat{a}_m^\dagger\hat{a}_m$. Therefore a generic Hamiltonian of the form

$$\hat{H}_{gen} = \sum_{n\neq m} h_{nm}\hat{c}_n^\dagger\hat{c}_m + g(\{\hat{n}_i\}) + f(t)\sum_l l\hat{n}_l, \tag{10}$$

will be transformed to

$$\tilde{H} = \sum_{n\neq m} e^{i\mathscr{A}(t)(n-m)}h_{nm}\hat{a}_n^\dagger\hat{a}_m + g(\{\hat{n}_i\}), \tag{11}$$

with time-dependence appearing only in the hopping term and with an arbitrary function $g(\{\hat{n}_i\})$ depending only on the number operators. For simplicity we will now restrict the discussion to a single band by setting $h_{nm} = \frac{J}{2}(\delta_{n,m-1} + \delta_{n,m+1})$. The total current is given by

$$\hat{J} = -i\frac{J}{2}\sum_n \left(e^{i\mathscr{A}(t)}\hat{a}_n^\dagger\hat{a}_{n+1} - e^{-i\mathscr{A}(t)}\hat{a}_{n+1}^\dagger\hat{a}_n\right), \tag{12}$$

which can be diagonalized to

$$\hat{J} = -J\sum_k \sin(k + \mathscr{A}(t))\hat{f}_k^\dagger\hat{f}_k, \tag{13}$$

where $\hat{f}_k = \sum_n \exp(ikn)\hat{a}_n/\sqrt{L}$. For $g(\{\hat{n}_i\}) = 0$, namely in the absence of interactions and external potentials, the Hamiltonian can be diagonalized simultaneously with $\hat{J}$, and therefore also the transformed Floquet operator:

$$\hat{U}(T,0) \equiv e^{-i\int_0^T d\bar{t}\,\tilde{H}(\bar{t})} = e^{-iJ\sum_k \hat{f}_k^\dagger\hat{f}_k \int_0^T d\bar{t}\,\cos(k+\mathscr{A}(\bar{t}))} \tag{14}$$

can be computed. Although the current is *not* conserved its expectation can be readily calculated:

$$\langle\hat{J}\rangle = -J\sum_k \sin(k + \mathscr{A}(t))\langle\hat{f}_k^\dagger\hat{f}_k\rangle. \tag{15}$$

Dynamical localization occurs when the total number of particles transported in one period vanishes, that is, when the integral of the current over one period yields zero for all $k$,

$$\int_{-T/2}^{T/2} dt e^{i\mathscr{A}(t)} = 0. \tag{16}$$

In this case the Floquet operator (14) reduces to the identity and its spectrum collapses such that all its eigenvalues are degenerate, making the stroboscopic dynamics trivial. Concretely, for the driving we use in this work (3), $\mathscr{A}(t) = A|t|$ (for $|t| < T/2$), and we have,

$$\int_{-T/2}^{T/2} dt e^{i\mathscr{A}|t|} = 2\int_{0}^{T/2} dt e^{iAt} = 2\frac{e^{iAT/2} - 1}{iA}, \tag{17}$$

which vanishes for $e^{iAT/2} = 1$, or $A/\omega = 2n \quad n \in \mathbb{Z}$.

When an external potential or interactions are present the Hamiltonian cannot be diagonalized in the basis of $\hat{J}$ and the above derivation does not apply since quasimomentum is not conserved. Notwithstanding, one might still expect some residual suppression of transport, since using the periodicity of $\exp[i\mathscr{A}(t)] = \sum_n C_n \exp[i\omega nt]$ one can cast the generic Hamiltonian (11) into the form

$$\tilde{H} = \sum_{n\neq m} C_0 h_{nm} \hat{c}_n^\dagger \hat{c}_m + g(\{\hat{n}_i\}) + \sum_{n\neq m} h_{nm} \hat{c}_n^\dagger \hat{c}_m \sum_s C_s e^{i\omega st}, \tag{18}$$

which includes a static Hamiltonian with a hopping term renormalized by $C_0$. This description is however oversimplified, due to the presence of a nontrivial driving term, which is coupled to the hopping. In the following Section we will explore what happens to transport in an interacting model, where the first term is set to vanish, namely $C_0 = 0$.

## 4 Results

We study density transport in a driven and interacting system. For this purpose we calculate the density-density correlation function

$$C_x(t) = \frac{1}{Z}\text{Tr}\left[\left(\hat{n}_x(t) - \frac{1}{2}\right)\left(\hat{n}_0 - \frac{1}{2}\right)\right], \tag{19}$$

which describes the spreading of a density excitation in the system. Here $Z$ is the dimension of the relevant Hilbert space, and the operators are written in the Heisenberg picture with respect to the *driven* Hamiltonian. Intuitively this describes the spreading of a density excitation in the presence of driving. Here, we fix the number of fermions to $N = (L-1)/2$, and use odd sizes $L$, such that an excitation at the center is at the same distance from the left and right (open) boundaries. To compute the excitation profile we stroboscopically evolve two initial states over $n$ periods, $|\psi_R(n)\rangle = \hat{U}(nT)\left(\hat{n}_0 - \frac{1}{2}\right)|\psi_0\rangle$ and $|\psi_L(n)\rangle = \hat{U}(nT)|\psi_0\rangle$, using a numerically exact method. The method is based on projecting the time evolution operator to an orthonormal basis of a truncated Krylov space spanned by the initial state, which is updated at each time step (for a pedagogical review of the method, see Sec. 5.1.2 of Ref. [12]). We then calculate the matrix element $C_x(nT) = \langle\psi_L(n)|\left(\hat{n}_x - \frac{1}{2}\right)|\psi_R(n)\rangle$. For the driving protocol we use in this work (3) the propagator $\hat{U}(nT)$ over $n$ periods is simplified to,

$$\hat{U}(nT) \equiv \mathscr{T}\exp\left[-i\int_0^{nT} \hat{H}(\bar{t})d\bar{t}\right] = \prod_{i=1}^{n}\left(e^{-i\hat{H}_- T/2}e^{-i\hat{H}_+ T/2}\right), \tag{20}$$

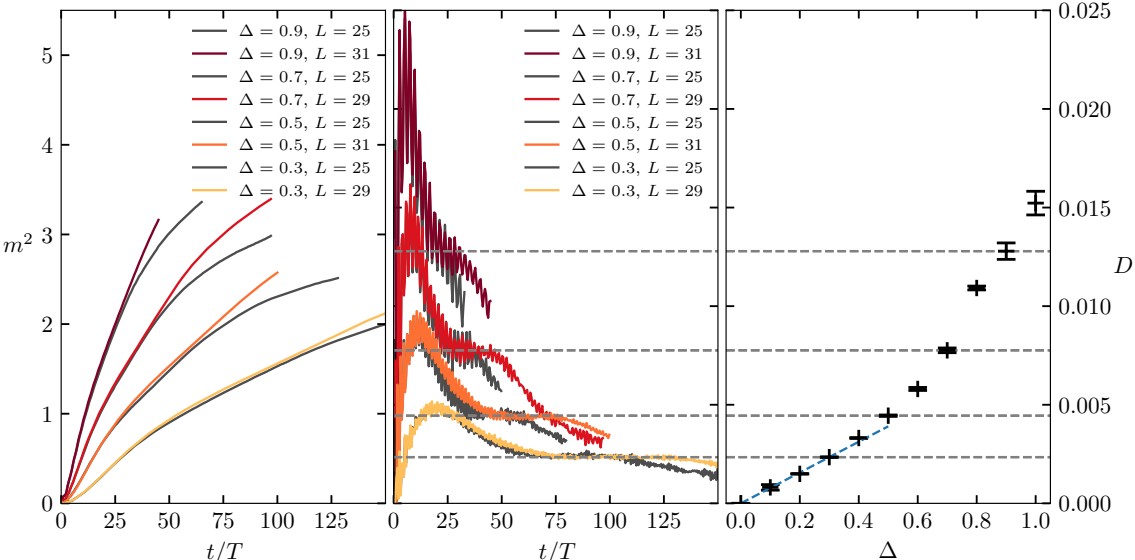

Figure 2: Mean-square displacement $m^2$ (cf. Eq. (21)) of the density excitation (left) and its derivative, the time-dependent diffusion coefficient $D(t)$, (middle) as a function of time for a few interaction strengths, $\Delta = 0.3, 0.6$ and $0.9$ (stronger interaction is indicated by more intense color). The colored lines indicate system sizes $L = 29$ and $L = 31$ and the grey lines correspond to $L = 25$, simulated with the same parameters. The horizontal dashed lines demonstrate the extraction of the diffusion coefficient from the flat part of the derivative, and the right panel shows the diffusion coefficient as a function of the interaction strength. The parameters used in this simulation are: $T = 5$ and the amplitude of the drive is set to the dynamical localization point $A_0 = 4\pi/T$ (23).

where $\hat{H}_\pm$ corresponds to the driven Hamiltonian (1) in the first and second halves of the period and $\mathcal{T}$ is the time ordering operator. The initial state $|\psi_0\rangle$ is sampled randomly from the Haar measure, which due to Lévy's Lemma approximates the trace in (19) with an error which is exponentially small in the system size [12, 34]. This computational scheme allows us to reach systems sizes as large as $L = 31$ sites, which corresponds to a Hilbert space dimension of $300, 540, 195$ (see Ref. [12] for more details on the numerical procedure).

To characterize the transport we calculate the mean-square displacement (MSD) of the density excitation,

$$m^2(t) \equiv \sum_x x^2 [C_x(t) - C_x(0)], \tag{21}$$

and the time-dependent diffusion coefficient [12, 35–37],

$$D(t) \equiv \frac{\mathrm{d}m^2}{\mathrm{d}t}. \tag{22}$$

We note in passing that within the linear response theory, namely when the drive is taken to be a small perturbation to the equilibrium state, this diffusion coefficient computed in the limit of $t \to \infty$ corresponds to the value calculated from the Kubo formula. In this work we however *do not consider the linear response regime*; our driving amplitudes are large and take the state of the system very far from equilibrium. Therefore there is no reason to expect that the diffusion coefficient calculated from (22) and the Kubo formula coincide. In fact it is not even clear *a priori* if transport in such cases will be diffusive. The dynamical localization of noninteracting systems, which was discussed in the previous Section, serves as an example when the diffusion coefficient (depending on the drive amplitude) is either zero or infinity and there is either no transport or the transport is ballistic.

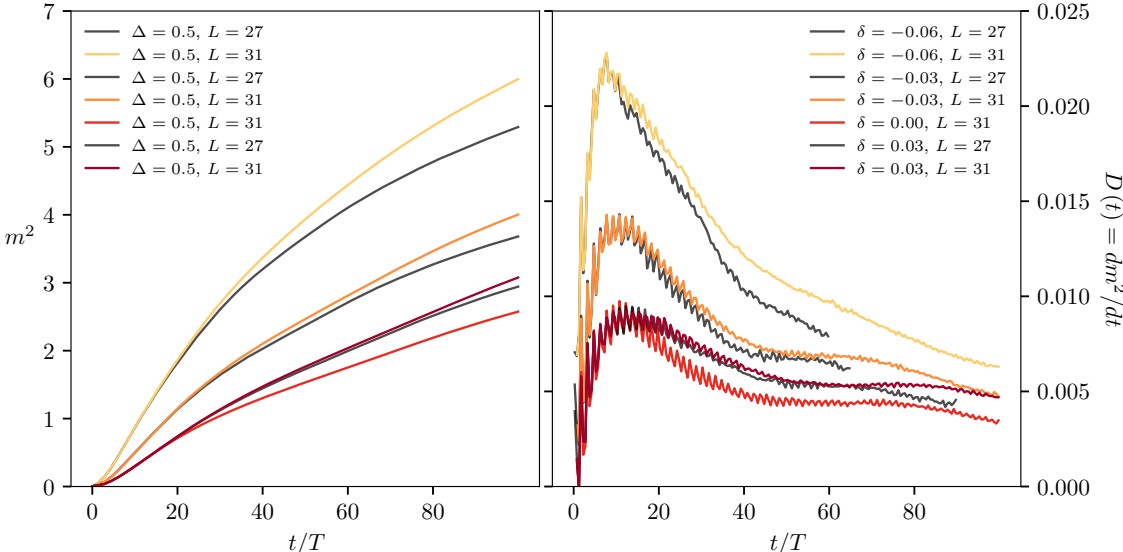

Figure 3: Same as Fig. 2, but detuning from the dynamical localization point by changing the amplitude of the drive, $\delta = 0, \pm 0.03$ and $-0.06$ (see (24)). The colored lines indicate system size $L = 31$ and the grey lines systems size $L = 27$, simulated with the same parameters. The interaction strength is fixed at $\Delta = 0.5$.

We now proceed to examine the stability of dynamical localization to the addition of interactions. To this end we set the system to the noninteracting dynamical localization point by setting the amplitude of the drive to

$$A_0/\omega = 2 \qquad A_0 = 4\pi/T. \qquad (23)$$

Throughout this work we fix the period of the drive to be $T = 5$, such that the frequency, $\omega = 2\pi/T \approx 1.256$ is smaller then the one-particle and many-body bandwidths, to allow effective energy distribution in the system using single and many-particle rearrangements. While the system is in the *noninteracting* dynamical localization point, the left panel of Fig. 1 shows sub-ballistic spreading of the density excitation, indicating that dynamical localization is unstable under the addition of interactions. The right panel shows fixed time cuts through the excitation profile. Interestingly, even after relatively long times these profiles do not approach a Gaussian form, suggesting that the density-density correlation function $C_x(t)$ is *not* well described by a diffusion equation. We also added an isoline at $C_x(t) = 0$ (the background of $C_x(t)$ is slightly negative due to the trivial correlation imposed by particle number conservation), which shows a peculiar feature at short times. We attribute this feature to short time transient dynamics which roughly corresponds to the location of the peak in the time dependent diffusion coefficient $D(t)$ shown in Fig. 2. The asymptotic transport is only reached at later times.

To characterize the nature of transport, and to estimate the finite size effects in the system we calculate the MSD (21) for different interaction strengths and several system sizes (not all shown for readability). From the left panel of Fig. 2 it is clear that while dynamical localization is destroyed by the addition of interactions, for weak interactions transport is still significantly suppressed and signalled by a slow growth of the MSD, which allows us to reach very long times before our results are affected by finite size effects. This time becomes shorter for stronger interactions, as could be inferred from the deviation in MSD between the various system sizes. After a short time of fast transient transport the MSD appears to enter a diffusive regime, as could be seen from a linear growth of the MSD. To see it more clearly we compute the time-

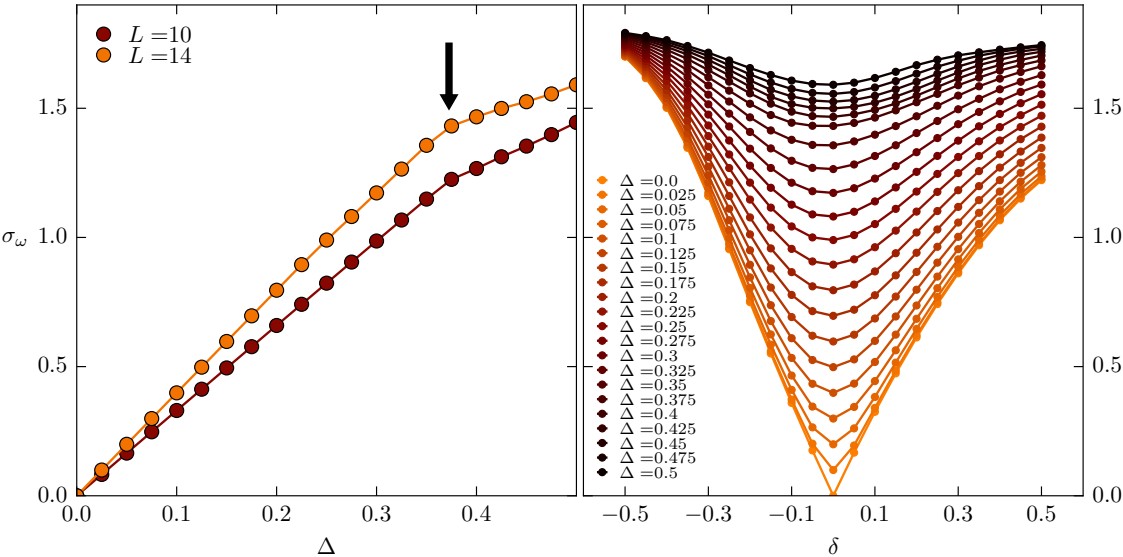

Figure 4: Left: Standard deviation $\sigma_\omega$ of quasienergies versus interaction strength, $\Delta$ for $\delta = 0$ (see (24)). Right: Standard deviation of quasienergies as a function of detunings $\delta$, for various interaction strengths, $\Delta$. The arrow indicates the interaction strength at which the quasienergy spectrum starts to wrap in the Floquet Brillouin zone, essentially at the same interaction strength for $L = 10$ and $L = 14$.

dependent diffusion coefficient (22) in the middle panel of Fig. 2. It shows a clear saturation to finite plateaus, which become longer for larger system sizes (while the height of the plateaus remains the same) and indicate that the departure from the plateau at later times is a finite size effect. The asymptotic diffusion coefficient is extracted from the height of the plateaus for different interaction strengths and is presented in the right panel of Fig. 2. While the overall dependence on the interaction strength is nonlinear, for sufficiently weak interaction strengths we find $D \sim \Delta$.

We now fix the interaction strength to $\Delta = 0.5$, such that we are able to reach a pronounced diffusive regime within our simulation times for $A/A_0 = 1$, and detune away from the dynamical localization point by slightly changing the amplitude of the drive. We characterize the strength of the detuning from the dynamical localization point by

$$\delta = \frac{A - A_0}{A_0}. \tag{24}$$

Fig. 3 shows the results for three different detuning strengths and two system sizes. Transport is much faster in this regime even for a few percent detuning from the dynamical localization point, requiring very large system sizes to capture the asymptotic transport. Both the MSD and the time-dependent diffusion coefficient show diffusive behavior for small detuning $\delta = \pm 0.03$, while for larger (in magnitude) detuning $\delta = -0.06$ it appears that our results do not capture the asymptotic transport even for the largest system size $L = 31$.

Finally we examine the broadening of the Floquet spectrum upon addition of interactions. As explained in Sec. 3, the spectrum of the operator $\hat{U}(T, 0)$ in (20) collapses to a single point at the noninteracting ($\Delta = 0$) dynamically localized point $\delta = 0$, resulting in a trivial stroboscopic dynamics. A perturbative reasoning suggests that a finite interaction will lift this massive degeneracy and broaden the quasienergy spectrum. We have confirmed this expectation by evaluating the quasienergies $\epsilon_\alpha$, defined as $\hat{U}(T, 0)|\alpha\rangle = \exp(-i\epsilon_\alpha T)|\alpha\rangle$, by exactly diagonalizing $\hat{U}(T, 0)$ up to systems sizes $L = 14$. The width of spectrum was calculated from

the standard deviation $\sigma_\omega$ of the $\epsilon_\alpha$ (after the quasienergies were shifted to the center of the band to avoid spurious widening due to wrapping around the Floquet Brillouin zone). The results are shown in Fig. 4, where we see that the spectrum broadens upon addition of interactions, however the location of the minimal (but still finite) width remains at zero detuning, $\delta = 0$ (see (24)). This is consistent with our observation that slowest transport occurs at this point (Fig. 3).

## 5  Discussion

In this work we have examined the stability of dynamical localization to the addition of interparticle interactions. We have found that while dynamical localization is destroyed by interactions, transport is significantly suppressed when the interacting system is at the noninteracting dynamically localized point (see right panel of Fig. 4 where the minimum of the width occurs at zero detuning, $\delta = 0$ for all considered interaction strengths $\Delta$). The slow transport allows us to find clear evidence of diffusive transport with a diffusion constant proportional to the interaction strength, indicating that scattering is dominated by interparticle collisions. We have also studied the dependence of the width of the quasienergy spectrum on the interaction strength. While the spectrum is completely degenerate for zero interactions, leading to trivial stroboscopic dynamics, for finite interaction the degeneracy is lifted and the width appears to be proportional to $\Delta$ before it saturates to its maximum value fixed by the driving frequency. From Fig. 4 one can see that for a *fixed* interaction strength the width increases with system size. This leads us to speculate that in the thermodynamic limit an arbitrarily weak interaction causes the spectrum to occupy the full allowed quasienergy range. While it is tempting to conclude that the width of the spectrum corresponds to the rate with which the dynamics unfolds, this is not true in general. For example, a finite width occurs for Floquet-MBL systems while the dynamics is frozen [15,16]. We therefore can only conclude that when the spectrum broadens dynamical localization may be destroyed, but the precise nature of the underlying dynamics has to be assessed by other means (as we have done in Figs. 2 and 3).

One interesting question on which our work might have some bearing is the locality of the effective Hamiltonian, $\hat{H}_F$. This quantity is defined from $\hat{U}(T,0)$ in (20) as $\hat{U}(T,0) = \exp\left(-i\hat{H}_F T\right)$ and is the generator of the stroboscopic time evolution. It has been speculated that for generic Floquet systems which are not MBL the effective Hamiltonian $\hat{H}_F$ is nonlocal [1, 38]. Since it is generally believed that spreading of correlations in systems with nonlocal *static* Hamiltonians will be superballistic, [39–42] one may wonder whether the diffusive transport we observe in this work points toward a *local* effective Hamiltonian. We would like to argue that due to the ambiguity in the definition of the effective Hamiltonian the question is not well posed. The ambiguity follows from the fact that $\hat{U}(T,0)$ and therefore all stroboscopically measured observables are invariant under the transformation $\hat{H}_F' = \hat{H}_F + \omega \sum_\alpha n_\alpha |\alpha\rangle\langle\alpha|$, where $|\alpha\rangle$ are the eigenstates of $\hat{U}(T,0)$ and $n_\alpha \in \mathbb{Z}$. Since in general the projectors $|\alpha\rangle\langle\alpha|$ are nonlocal objects this renders the locality of $\hat{H}_F$ a property which cannot be inferred from physical observables. While we believe that the discussion of locality of $\hat{H}_F$ is irrelevant as far as physical observables are concerned, it remains an interesting question whether the ultimate fate of driven interacting systems is encoded in the locality of the corresponding *Floquet operator,* $\hat{U}(T,0)$, which *is* a physical quantity and whose locality can be assessed, for example using the operator entanglement entropy [43]. We leave this question for future work.

## Acknowledgments

YB acknowledges funding from the Simons Foundation (#454951, David R. Reichman). Our code uses the libraries PETSc [44–46] and SLEPc [47]. This work used the Extreme Science and Engineering Discovery Environment (XSEDE), which is supported by National Science Foundation grant number ACI-1548562. This project has received funding from the European Union's Horizon 2020 research and innovation programme under the Marie Skłodowska-Curie grant agreement No. 747914 (QMBDyn). DJL acknowledges PRACE for awarding access to HLRS's Hazel Hen computer based in Stuttgart, Germany under grant number 2016153659.

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
