# Peer review of "Absence of dynamical localization in interacting driven systems"

_SciPost Physics, doi:SciPost Phys. 3, 029 (2017)_

## Round 1 · Referee Report · Anonymous · 2017-8-30

Strengths
1. Interesting and timely problem.
2. Numerics are very impressive.
Weaknesses
1. Lacking details on methods.
2. Could argue some of the assertions better. Especially the finite size effect on diffusion saturation.
Report
The manuscript "Absence of dynamical localization in interacting driven systems" is an interesting study of dynamics of driven interacting systems. It asks whether a system whose single-particle orbitals are localized by an oscillating force can delocalize due to interactions. It answers this question by considering the spreading of the particle densities in large chains, while sampling over all possible initial wave functions. The authors identify a diffusive era in the evolution of the system, and show that the effective diffusion constant for this era is linear in the interaction strength. The results appear reasonable and interesting.
There are a couple of things that the authors could do to improve the manucript. First, there is a strong claim that the diffusive era would extend indefinitely had it not been for the finite size of the system. It would be nice to support this claim with a back of the envelope calculation on whether the length and location of the saturated diffusion era matches the expected diffusion scaling. Naively, I would expect that $latex \sqrt{D t_{max}}$ would be of the order of the size of the system, with $latex t_{max}$ being the time at the end of the diffusive era. Given the fast motion era preceding the diffusion era, such arguments require a tad more care. Needless to say that having more data on shorter system and some scaling collapse would be even more appealing.
Also, the numerics is quite impressive, and simulates a chain up to 31 sites long. The authors mention that the numerics follows a Krylov-based method (bottom of page 6), but do not cite or elaborate. The reader would greatly benefit from more information about the numerical calculation, either through a citation, or, even better, through an appendix.
Lastly, figure 3 is hard to read due to the many colors and the relatively small color bars. Putting the labels near the lines could be more effective.
Requested changes
1. More details on numerics.
2. Finite size scaling study of the diffusion plateu.
Author: David J. Luitz on 2017-09-28 [id 174]
(in reply to Report 1 on 2017-08-30)
We are grateful to the referee for the positive evaluation of our work. Below we address in detail the comments raised in the review report.
1) The details of the numerical scheme are published in Sec 5.1.2 of Ref. [12]. We have rewritten the sentences concerning the description of the method to be more clear and have pointed to Ref [12], Sec 5.1.2 explicitly. 2) We have improved the presentation of Fig. 2 and 3 by increasing the size of the legend, facilitating the distinction between the different lines. 3) The referee has suggested that a proper finite size scaling analysis of the diffusion plateau would further reinforce the claim of asymptotic diffusion. In the manuscript we have already shown a systematic increase in the length of the plateau for longer system sizes, which suggests that the observed diffusion in the thermodynamic limit might be asymptotic. We agree with the referee that in principle from the diffusive behavior a naive finite size scaling of the plateau length can be expected. However, given the available system sizes and the difficulty of determining the exact location of departure from the plateau, this can not convincingly be seen. Moreover, we note that the shape of the excitation is strongly non-gaussian (cf Fig 1) and therefore corrections to the finite size scaling are expected, since the tail of the excitation reaches the boundary earlier, leading to significant deviations from the naive expectation. We believe that the computational effort required to perform a more quantitative analysis would not be on par with the added benefit. Smaller systems are not very useful here due to the nearby crossover from ballistic transport mentioned by the referee.
Author: David J. Luitz on 2017-09-28 [id 175]
(in reply to Report 2 on 2017-09-27)We thank the referee for contributing this report. We comment on the critique drawn from the report below:
1) While it is very possible that diffusion is “universal” near dynamical localization points it is not clear to us how to define the appropriate “universality class”. Clearly this has to be done analytically, since numerically one can only study a handful of specific systems. While we agree with the referee that the destruction of dynamical localization by interaction is not very surprising, the diffusive transport we observe far-from linear response is. It suggests that there might be a generalization of linear-response theory tailored for far-from equilibrium systems. We however left this fascinating direction for future work. 2) We have added a colorbar to Fig. 1, and indicated the plotted quantity in the caption. 3) We have decided not to include the used shift in the right panel, since we feel it adds an unneeded detail and makes the caption cumbersome. The reason for this is that the shift is based on a completely unphysical and rather visual quantity, namely the number (10) of shown lines in the plot, leading to an equidistant spacing. We think that the baseline for each curve is easy enough to extract from the plot. 4) We have added a citation to the definition of MSD. 5) We corrected various typos pointed out by the referee. 6) We explicitly noted that the expression for A(t) is only valid in the “first” period. 7) Please see reply to Referee 1 on the finite size effects in the system. We have checked carefully using a set of system sizes ranging from L=19...31 that the diffusive plateau increases systematically with system size, which is also visible in the plots shown in the paper. For readability of the figures, we show however only two system sizes for each interaction strength and detuning. 8) The last sentence of the abstract was rewritten.

---

## Round 1 · Referee Report · Anonymous · 2017-9-27

Strengths
The manuscript is clearly written, and the numerical results seem consistent. The authors use a clever numerical method allowing them a good generic characterization of the dynamics of relatively large systems by propagating only two vectors in time.
Weaknesses
1) The basis for several assertions need to elaborated and backed with numerical data
2) Is there any universal behavior near the "dynamical localization point"? If so, it would make the results much more interesting. However, the authors do not give any indication that such behavior indeed occurs.
Report
The paper by D.J. Luitz et al investigates how interactions affect transport in a system which is dynamically localized in the non-interacting case. The authors find that the addition of interactions makes transport diffusive even at the dynamical localization point (after a transient period of fast transport). While the diffusion coefficient vanishes at the non-interacting dynamical localization point, it increases continuously when either interactions are increased, or the driving amplitude is tuned away from this special point. The authors conclude with an interesting speculation that the sub-ballistic spreading of correlations might indicate that the effective Hamiltonian is local, in contrast to the generic expectation from non-MBL Floquet systems. Overall, the authors performed a thorough job at studying the system, including rather impressive numerics, and their findings are reasonable (although not very surprising. ). I recommend this paper for publication in SciPost, once the points below are addressed.
Requested changes
. I therefore believe that the paper is suitable for publication in SciPost, once the following points are addressed:
The authors may wish to consider the following minor comments for better readability:
- Fig. 1: Left panel: It would be helpful to explicitly indicate on the figure what quantity is plotted and present a colorbar. Right panel: the authors should indicate the background value (e.g. for t/T=0.5) and the shift for the other values of t/T? Also, could be helpful to remove y-axis labels and ticks beyond 10^0. Caption: would be helpful to define C_x(t), or to refer the reader to eq. 19.
- Fig. 2 caption: would be helpful to define the MSD or refer the reader to its definition. 4th line - seems to be missing "that".
- Text: results pg. 6 "the spreading of density excitation" - missing 's'. Eq. 20: middle part missing time ordering. Discussion pg. 10 line 5 - there's an extra parenthesis.
- Eq. (10) I would recommend that the authors change the index “k” to something else, to avoid confusion.
- Above Eq. (17), the authors give the form for A(t). Is this form valid only for –T/2 to T/2 ?
- The numerical results for the diffusion constant show deviation from a platuea at late times. The authors attribute this to finite size effects. Do they have numerical evidence supporting this assertion? I recommend that the authors elaborate more on this statement.
- I found the last sentence of the abstract hard to parse, and I recommend that the authors rewrite it.

---

## Editorial Decision

published